# Repercussions of the Calpain Cleavage-Related Missense Mutations in the Cytosolic Domains of Human Integrin-β Subunits on the Calpain–Integrin Signaling Axis

**DOI:** 10.3390/ijms26094246

**Published:** 2025-04-29

**Authors:** Reshma V. Kizhakethil, Ashok K. Varma, Sagar H. Barage, Neelmegam Ramesh Kumar, Kayalvizhi Nagarajan, Aruni Wilson Santhosh Kumar, Shashank S. Kamble

**Affiliations:** 1Amity Institute of Biotechnology, Amity University, Mumbai 410206, Maharashtra, India; reshmavk90@gmail.com (R.V.K.); sagarbarage@gmail.com (S.H.B.); neelamegamramesh@gmail.com (N.R.K.); vcamitymumbai@gmail.com (A.W.S.K.); 2Advanced Centre for Treatment, Research and Education in Cancer, Kharghar, Navi Mumbai 410210, Maharashtra, India; avarma@actrec.gov.in; 3Homi Bhabha National Institute, Training School Complex, Anushakti Nagar, Mumbai 400094, Maharashtra, India; 4Centre for Computational Biology and Translational Research, Amity University, Mumbai 410206, Maharashtra, India; 5Department of Zoology, Periyar University, Periyar Palkalai Nagar, Salem 636011, Tamil Nadu, India; kayalvizhinagarajan@gmail.com; 6California University of Science and Medicine, Colton, CA 92324, USA; 7Centre for Drug Discovery and Development, Amity University, Mumbai 410206, Maharashtra, India

**Keywords:** calpain cleavage-related mutations, calpain, integrin, tumor metastasis, uterine corpus endometrial carcinoma

## Abstract

Calpains, calcium-dependent cytosolic cysteine proteases, perform controlled proteolysis of their substrates for various cellular and physiological activities. In different cancers, missense mutations accumulate in the genes coding for the calpain cleavage sites in various calpain substrates termed as the calpain cleavage-related mutations (CCRMs). However, the impact of such CCRMs on the calpain–substrate interaction is yet to be explored. This study focuses on the interaction of wild-type and mutant β-integrins with calpain-1 and 2 in uterine corpus endometrial carcinoma (UCEC). A total of 48 calpain substrates with 176 CCRMs were retrieved from different datasets and shortlisted on the basis of their involvement in cancer pathways. Finally, three calpain substrates, ITGB1, ITGB3, and ITGB7, were selected to assess the structural changes due to CCRMs. These CCRMs were observed towards the C-terminal of the cytoplasmic domain within the calpain cleavage site. The wild-type and mutant proteins were docked with calpain-1 and 2, followed by molecular simulation. The interaction between mutant substrates and calpains showcased variations compared to their respective wild-type counterparts. This may be attributed to mutations in the calpain cleavage sites, highlighting the importance of the cytoplasmic domain of β-integrins in the interactions with calpains and subsequent cellular signaling. Highlights: 1. Calpain cleavage-related mutations (CCRMs) can alter cellular signaling. 2. CCRMs impact the structure of C-domains of human integrin-β subunits. 3. Altered structure influences the cleavability of human integrin-β subunits by human calpains. 4. Altered cleavability impacts the cell signaling mediated through calpain–integrin-β axis. 5. Presence of CCRMS may influence the progression of uterine corpus endometrial carcinoma (UCEC).

## 1. Introduction

Calpains are cytosolic cysteine proteases that require calcium for their activation. They are involved in various physiological activities, such as cell signaling, motility, ion transport, apoptosis, and cancer progression [1,2,3]. Calpain influences cancer progression by mediating the proteolytic cleavage of its substrates, which are involved in the tumor invasion and metastasis pathways (Rios-Doria et al., 2003) [4,5,6,7]. Under different physiological conditions, calpains can lead to cellular deadhesion and cytoskeletal remodeling, which are essential for endocytosis and cellular migration (Storr et al., 2011) [8]. Calpain reportedly plays a role in weakening the intracellular links to integrins, eventually leading to their detachment from the cell (Perrin and Huttenlocher, 2002) [9]. Calpain activity is regulated through the calcium ion concentration and post-translational modifications (Chen et al., 2020) [10,11,12]. Calpain constitutes a family of 16 members, among which calpain-1 and calpain-2 are the most extensively studied. The calpain family also consists of an endogenous inhibitor, calpastatin [4,13,14], which, under different physiological conditions, tightly regulates calpain–substrate interactions (Yadavalli et al., 2004) [15]; however, dysregulation of the activities of different calpains is observed in various cancers, probably through the downregulation of calpastatin (Zhang et al., 2019) [16]. Individual calpains, such as calpain-3, calpain-5, calpain-6, and calpain-10 [17,18,19]^,^ also play distinct roles in numerous cellular activities.

Calpains function by cleaving their protein substrates at specific locations, which leads to a cascade of cellular signaling events. Calpain aberrations are also observed in different pathological conditions (Penna et al., 2008) [13,20,21,22,23] because of the enormous repertoire of calpain substrates and their involvement in distinct cellular activities. The presence of various genetic mutations is one of the established hallmarks of all types of cancers. Mutations in the functional domains of calpains have been reported to perturb protein–protein interaction networks and downstream signaling pathways [24,25].

Human integrins are transmembrane heterodimeric glycoproteins crucial for cell–cell and cell–matrix interactions. Glycosylation of integrin beta subunits has been shown to play a key role in maintaining the active conformation required for ligand binding and subsequent downstream signaling pathways. The degree and type of glycosylation of integrin beta subunits have been shown to change during cancer progression (Cai et al., 2017; Pocheć et al., 2003) [5,26]. Structurally, each integrin consists of non-covalently linked α and β subunits, which together form a large extracellular domain responsible for ligand binding and specificity. Integrins are a diverse family of adhesion and signaling receptors crucial for various physiological processes. They interact with a wide range of extracellular and intracellular proteins through different sequence motifs. These interactions play a significant role in regulating important biological processes, such as angiogenesis, homeostasis, and immune cell function. Dysregulation of integrin function can contribute to various pathological conditions, including cancer metastasis, inflammatory diseases, and vascular disorders (Mezu-Ndubuisi and Maheshwari, 2021) [27].

The protein–protein interaction between calpain and the integrin-β subunit plays a crucial role in various cellular functions, impacting processes such as cell migration, adhesion, and signaling pathways (Yadavalli et al., 2004) [15]. Therefore, the proposed study focused on missense mutations in the cytosolic domains of the integrin-β subunits identified and retrieved in the case of uterine corpus endometrial carcinoma (UCEC) (Figure 1). Several studies have attempted to evaluate the substrate specificity of calpain with respect to the outcome of calpain–substrate interactions [2,5,28]. Therefore, the impact of mutations in the calpain cleavage sites within the calpain substrates on the calpain–substrate interaction and downstream signaling pathways needs to be explored.

## 2. Results

### 2.1. Data Retrieval

A total of 48 calpain substrates and 176 missense mutations in UCEC were analyzed for their functional significance associated with protein–protein interactions in tumorigenesis. Three calpain substrates, ITGB1, ITGB3, and ITGB7, were subsequently selected for further study on the basis of their importance in tumor invasion and metastasis as well as the occurrence of missense mutations within the calpain cleavage site.

### 2.2. Effects of Mutations on Pathogenicity and Calpain Cleavage

The pathogenicity of the T777M, R760H, R786Q, and D749N mutations was confirmed via different pathogenicity tools (Table 1). These scores reflect the effects of each of these mutations on cancer pathophysiology, as mentioned in The Cancer Genome Atlas database. Furthermore, the effects of mutations in the calpain cleavage sites of these substrates were predicted via DeepCalpain (Table 2). The wild-type ITGB1 gene had five calpain cleavage sites, which were reduced to three in the T777M mutant, with the loss of two sites at positions 778–779 and 788–789. Similarly, the wild-type ITGB7 had six calpain cleavage sites, which were reduced to five in the ITGB7 mutant, with a loss of the site at positions 769–770. However, the ITGB3 wild-type and the D749N mutant presented no loss of the calpain cleavage sites, whereas the second mutant, R786Q, presented ten sites with the loss of two calpain cleavage sites at positions 772–773 and 779–780. The variations in the number of calpain cleavage sites between the wild-type and mutant substrates reveal an alteration in the cleavage pattern of the integrin-β subunit cytosolic domain, which further needs to be validated via in vitro studies.

### 2.3. Modeling and Simulation

The full-length structures of wild-type ITGB1, ITGB3, and ITGB7 were retrieved from AlphaFold, and mutations at specific positions were introduced via PyMOL (Schrodinger, L. (2010) The PyMOL Molecular Graphics System, Version 1.3r1., n.d.) [29]. The wild-type and mutant structures were subjected to modeling and simulation on Gromacs 2021.2 for better conformational stability. The root mean square deviation (RMSD), root mean square fluctuation (RMSF), and radius of gyration (Rg) of the wild-type and mutant structures were analyzed. In the initial time step, the variation in the RMSD between wild-type and mutant ITGB1 was very small. At approximately 20 ns, the difference in RMSD was not as evident at 1.4 nm for the ITGB1 wild-type (black line) and at 1.2 nm for the ITGB1 mutant (red line). A drastic change in the RMSD was observed at 35 ns with wild-type ITGB1, ~1 nm, and the ITGB1 mutant at ~1.75 nm. A slight variation between wild-type and mutant ITGB1 continued from 60 ns to 90 ns, eventually ending with a minimum variation in wild-type ITGB1, which was ~1.4 nm, and the ITGB1 mutant, which was ~1.24 nm (Figure 2a). The RMSF of wild-type and mutant ITGB1 aligned throughout, starting from 1 nm, but with very little variation between residues 180 and 320. However, a major variation was observed between residues 750 and 800, with ITGB1 wild-type falling at ~1.25 nm and the ITGB1 mutant at ~0.3 nm, indicating the impact of the mutation at position 777 (Figure 2b). The radius of gyration for wild-type and mutant ITGB1 in the initial time steps aligned at 4.5–4.7 nm for 3 ns, followed by variation throughout. The greatest difference was observed between 30 and 40 ns, with ITGB1 wild-type touching a value of ~5 nm and the ITGB1 mutant touching a value of 3.81 nm (Figure 2c).

In ITGB7, greater variation was observed in the RMSD, RMSF, and radius of gyration values between the wild-type and mutant strains. For the initial time step of 3 ns, the RMSD of the wild-type and mutant strains aligned completely, followed by slight variation. After 15 ns, the graph shows a shift in the values of the wild-type and mutant strains up to 100 ns. The ITGB7 wild-type (black line) fell within the range of 1.5–2 nm from 15 ns to 100 ns, whereas the ITGB7 mutant (red line) presented greater variation, with the highest peak reaching ~4.5 nm at approximately 30 ns and 70 ns and the lowest values hitting approximately 2–2.3 nm from 15 ns to 100 ns (Figure 3a). The RMSF numbers between the ITGB7 wild-type and mutant structures did not align, with an average difference of approximately 1 nm observed throughout the simulation. However, the differences between wild-type and mutant ITGB7 in the cytosolic domain (700–800 amino acids) were greater as a result of the mutation at position 760 (Figure 3b). The radius of gyration also shows a similar pattern between the wild-type and mutant structures, with both aligning in the initial time frame of 1 ns. The ITGB7 wild-type then shifted from 4.25 nm at approximately 1 ns to 3.5 nm at 100 ns, whereas the ITGB7 mutant peaks slightly fluctuated throughout at different time intervals, with the highest being 5 nm at ~37 ns and the lowest being 4.1 nm at ~9 ns (Figure 3c).

In ITGB3, the two mutants D749N and R786Q were compared with the ITGB3 wild-type structure. Initially, RMSD values of the wild-type and mutant structures followed a similar pattern until 40 ns, with values ranging between 0.25 nm and 2 nm. At approximately 40 ns, the ITGB3 wild-type (black line) depicted a sudden dip from 2 nm to 0.75 nm, which continued up to 100 ns with slight variations. After 40 ns, the RMSD value of the D749N mutant (red line) slightly increased to 2.5 nm and continued to increase till 100 ns. Whereas the RMSD value of the R786Q mutant (green line) continued to increase to approximately 2 nm until 60 ns and then decreased to 1 nm, followed by an increase to approximately 2.5 nm between 90 and 100 ns (Figure 4a). The RMSF values of wild-type ITGB3 initially ranged from 0.5 nm to 1.75 nm and continued to range from 0.25 nm to 1 nm until the 600th residue. The RMSF value of the ITGB3 R786Q mutant initially ranged from 1 to 1.2 nm, followed by significant changes between the residues from 600 to 750, with a spike reaching a value of 2 nm, whereas that of the wild-type protein remained at 0.25 nm. In the ITGB3 D749N mutant, the initial RMSF values were less than 1 nm and showed slight variation between the residues from 700 to 800 compared with those of the wild-type, whereas in the R786Q mutant, the initial RMSF values were ~1.25 nm, with a major variation from 600 to 750 nm and a peak at 2 nm in the region (Figure 4b). The D749N and R786Q mutants showed greater variation in the region of amino acid residues 600–750, with R786Q having a peak at 2 nm, whereas D749N was in the region of 0.5–1 nm, followed by very slight variation in the 750–800 residue region. The radius of gyration for the wild-type and mutant structures was ~4.75 nm and continued to be in the range of 4–4.75 nm until 30 ns, with little fluctuation. After 30 ns, wild-type ITGB3 increased from 4 nm to 4.75 nm, followed by slight fluctuations with few peaks. The D749N mutant showed a slight dip at 40 ns and continued to be in the range of 3.5–3.75 nm with very few fluctuations, whereas the R786Q mutant showed larger fluctuations and peaks thereafter up to 100 ns (Figure 4c).

Overall, ITGB1, the T777M mutant, presented a slightly greater RMSD from the wild-type structure, indicating that conformational alterations occurred during the course of the simulation. Additionally, wild-type and mutant ITGB1 fluctuated less throughout the protein length except in the last 100 amino acid residues, as indicated by the trend in RMSF. ITGB1 wild-type, with an average value of 4.72 nm, showed a less compact structure than the mutant, with an average value of 4.19 nm, as indicated by the radius of gyration. Compared with the wild-type structure, the R760H mutant of ITGB7 presented a greater RMSD, indicating the presence of a wider conformational space, whereas the wild-type structure continued to exhibit very few fluctuations. The radius of gyration of the ITGB7 mutant displayed a very high average value of 4.6 nm compared with that of the wild-type, with an average value of 3.72 nm, indicating a less compact structure. In ITGB3, the D749N and R786Q mutants showed significant fluctuations from the wild-type structures, indicating greater conformational changes and greater flexibility. Compared with those of the wild-type ITGB3, the average radii of gyration values of the D749N and R786Q mutants were 4.38, 4.07, and 4.56 nm, respectively, indicating that the D749N mutant is more compact and the R786Q mutant is less compact.

### 2.4. Molecular Docking

The ITGB1, ITGB3, and ITGB7 wild-type and mutant substrates were docked with the respective calpains via the HDock server. The preferred docking sites of both the substrates and ligand were specified for targeted docking. Among the various docked poses, the best-docked poses of ITGB1, ITGB3, and ITGB7, both wild-type and mutant, were then selected on the basis of the maximum number of amino acid residues from the calpain cleavage site involved in the interaction with calpain, and their docked scores were compared. The intermolecular interactions of the docked complexes were further studied in Discovery Studio. In ITGB1 (Figure 5a,b), the region comprising 765-798 residues (34 residues) was observed to be the calpain cleavage site, of which 14 residues were involved in the interaction with calpain-2 in the wild-type strain, whereas 17 residues in the mutant strain were at the interface with calpain-2. Compared with the mutant, the wild-type protein was more stable, with a docking score of −222.61, suggesting that the mutation might have affected the interaction of ITGB1 with calpain-2. Compared with those in the ITGB1 wild-type and calpain-2 complex, the number of hydrogen bonds between the ITGB1 mutant and calpain-2 complex decreased by 4, suggesting a weaker interaction. There were also reductions in hydrophobic and electrostatic interactions in the mutant docked complex compared with those in the wild-type complex (Table 3). In a similar way, the interaction of ITGB7, wild-type and mutant structures with calpain-2 was studied (Figure 6a,b). The region spanning the residues 740–798 (59 residues) was observed to be the calpain cleavage site in ITGB7. Among these residues, 15 residues of the calpain cleavage site in the wild-type protein and 14 in the mutant protein interacted with calpain-2, whereas both the wild-type ITGB7–calpain-2 docked complex and the mutant ITGB7–calpain-2 docked complex exhibited approximately the same docking scores of −222.31 and −222.50, respectively, suggesting minimal impact of the mutation under study on the interaction with calpain-2. However, intermolecular interactions revealed a decrease in the number of hydrogen bonds and hydrophobic and electrostatic interactions between the ITGB7 mutant and calpain-2 compared with those between the ITGB7 wild-type and calpain-2 (Table 4). ITGB3 is another substrate of calpain-1 (Figure 7a,c), where the residues between 745 and 788 (44 residues) were observed to be the calpain cleavage site. Out of these, 22 residues from the calpain cleavage site of wild-type ITGB3, 18 residues from the D749N mutant, and 22 residues from the R786Q mutant interacted with calpain-1, with docking scores of −283.58, −234.95, and −224.14, respectively. Compared with those in wild-type ITGB3, the number of hydrogen bonds in the mutants and their hydrophobic and electrostatic interactions were lower (Table 5). The reduction in the intermolecular bonding of the docked complexes of ITGB1, ITGB7, and the ITGB3 mutants compared with their wild-type counterparts indicates a change in interaction that can be attributed to the CCRMs in these protein substrates. In the docked complexes of the calpain–integrin-β subunit, the interface residues were slightly changed in the mutant protein substrates ITGB1, ITGB7, and ITGB3 compared with those in the wild-type protein (Table 6a–c). Overall, the changes observed in the docking score and number of hydrogen bonds in the intermolecular interactions of wild-type and mutant ITGB1, ITGB3, and ITGB7 might affect the binding affinity, which needs to be further validated with in vitro studies.

### 2.5. Molecular Dynamics Simulation of the Docked Complexes

To further validate the docking results, molecular dynamics (MDs) simulations were performed on seven different complexes: (1) ITGB1 wild-type–calpain-2; (2) ITGB1 mutant–calpain-2; (3) ITGB7 wild-type–calpain-2; (4) ITGB7 mutant–calpain-2; (5) ITGB3 wild-type–calpain-1; (6) ITGB3 mutant (D749N)–calpain-1; and (7) ITGB3 mutant (R786Q)–calpain-1. These simulations aimed to study the structural and dynamic behaviors of these proteins in their wild-type and mutant forms and to understand the effects of specific mutations on their stability and interactions.

In the RMSD trajectories of the ITGB1–calpain-2 complexes, during the initial timeframe of 0-5 ns, the wild-type (black) complex and mutant (red) complex were in the range of 0.15 nm–0.7 nm and continued to slightly fluctuate until 30 ns in the range of 0.3–0.7 nm (Figure 8a). From ~30 ns to 40 ns, the RMSD values of the wild-type complex ranged from 0.5 to 0.75 nm, and that of the mutant complex ranged from 0.45 to 0.65 nm. Compared with the mutant complex, the wild-type complex showed very minor fluctuations from 40 ns to 100 ns, ranging from 0.55 to 0.7 nm, whereas the mutant complex displayed more fluctuations, with the peak reaching 0.75 nm at ~50 ns and decreasing to 0.4 nm at 60 ns, followed by an increase to 1.15 nm at ~70 ns. From 70 ns to 100 ns, the RMSD values of the mutant complex fluctuated between 0.7 and 1.2 nm. The overall RMSD analysis indicated possible structural adjustments or reorientations within the binding pocket, possibly due to changes in the mutant receptor. Additionally, a detailed analysis of the root mean square fluctuation (RMSF) of each residue over the course of the simulation was performed (Figure 8b). Both the wild-type (black) and mutant (green) forms exhibited increased fluctuations in certain residues, likely indicating flexible regions or loops. The mutant ITGB1 showed more pronounced peaks, indicating that specific regions experienced greater instability and movement. A similar observation was noted for ligands where calpain bound to the mutant receptor had greater fluctuations in the residues towards the end terminal of the protein, as shown in the blue color of the RMSF plot. The radius of gyration (Rg) for the ITGB1–calpain-2 complexes indicated differences in the compactness of the docked complex structures (Figure 8c). The wild-type complex (black) started at ~4.4 nm at 0 ns and ended at 4.15 nm at 100 ns, remaining in the same range throughout, whereas the mutant (red) complex started at ~5.15 nm at 0 ns and ended at 5.4 nm at 100 ns, being in the range of 5–5.4 nm throughout the simulation. This finding revealed that the wild-type complex was more compact than the mutant complex in ITGB1. Hydrogen bond analysis was performed for both the complexes between the receptor (ITGB1) and ligand (Calpain-2) over a 100 ns time frame (Figure 8d). The wild-type (black) complex maintained a greater number of hydrogen bonds throughout the simulation, which is indicative of stable binding interactions. The mutant (red) complex showed a reduced number of hydrogen bonds, suggesting weakened or altered binding interactions due to mutations.

The RMSD values of the ITGB7–calpain-2 complex in the wild-type (black) and mutant (red) conformations started at ~0.15 nm, and that in the wild-type complex continued to increase very gradually, with very minor fluctuations until 40 ns (Figure 9a). Between 40 and 60 ns, slightly more fluctuations can be observed, with the lowest point of ~0.3 nm at approximately 45–50 ns and the highest peak of ~0.6 nm at ~60 ns. From 60 to 100 ns, the wild-type ITGB7–calpain-2 complex continued to fluctuate, with the highest peak at ~0.75 nm at 75 ns and the lowest peak at 0.45 nm at 80 ns and 97 ns. At the same time, the mutant ITGB7–calpain-2 complex was observed to have greater fluctuations during the initial time frame, with a sudden rise to 0.7 nm at ~4 ns and again falling to 0.4 nm at ~6 ns. Furthermore, a sudden increase was observed at 0.8 nm close to 10 ns, and the next peak was observed at ~12 ns to 0.89 nm. This was followed by a sudden decrease to 0.45 nm at approximately 16–17 ns and a rise to ~0.75 nm at 20 ns, after which it continued with comparatively fewer fluctuations than the initial time frame. From 20 to 100 ns, the RMSD values of the mutant continued to range from 0.75 to 1.0 nm, with the lowest value at ~30 ns and the highest value at ~90 ns, with a gradual increase between 20 ns and 100 ns. Compared with the mutant, the wild-type ITGB7–calpain-2 complex displayed better structural integrity throughout the simulation. The RMSF plot highlighted the flexibility of individual residues in the ITGB7 proteins (Figure 9b). For the wild-type receptor (black) and ligand (red), lower RMSF values suggest stable regions with minimal flexibility, indicating strong interaction and structural consistency. However, the mutant ITGB7 (green)–calpain-2 (blue) complex showed significant fluctuations in specific residues (indicated by the pronounced peaks in the green line). These fluctuations suggest increased flexibility and instability in certain regions, likely due to the impact of the CCRMs on the protein’s secondary structure. The higher RMSF values towards the C-terminal regions of the mutant ITGB7–calpain-2 complex indicated areas of increased mobility, potentially affecting binding interactions and overall stability. The radius of gyration (Rg) for the wild-type (black) and mutant ITGB7 (red)–calpain-2 complexes indicated differences in the compactness of the docked complex structures (Figure 9c). The wild-type complex starting at 3.85 nm continues to be in the range of 3.8–4.05 nm throughout the simulation, with few fluctuations. The mutant complex started at 4.55 nm, reached 4.75 nm at ~7 ns, and then gradually decreased to 4.25 nm at 20 ns. From 20 ns to 100 ns, it remained in the range of 4.25–4.45 nm throughout the simulations, with slight fluctuations. Overall, the wild-type ITGB7–calpain-2 complex was found to be more compact and stable than the mutant ITGB7–calpain-2 complex. The hydrogen bond plot provided insights into the strength of the interactions between ITGB7 and calpain-2 (Figure 9d). The wild-type ITGB7–calpain-2 complex (black) formed a moderate number of hydrogen bonds throughout the simulation, indicating stable binding interactions. Interestingly, the mutant ITGB7–calpain-2 complex (red) showed an even greater number of hydrogen bonds, suggesting compensatory interactions due to mutation-induced structural changes. However, increased hydrogen bonding does not necessarily translate to stronger binding affinity, as the quality and distribution of these bonds are critical for functional stability. The greater number of hydrogen bonds in the mutant ITGB7 form may indicate attempts to stabilize the altered conformation, but the overall impact on binding affinity remains complex.

With respect to the RMSD of the ITGB3–calpain-1 complex, the wild-type (black) and D749N (red) and R786Q (green) mutants initiated at 0.25 nm (Figure 10a). The RMSD of the wild-type complex gradually shifted to 1.5 nm at ~10 ns, followed by a decrease to ~0.5 nm at 20 ns. The RMSD then increased again to 2 nm at ~30 ns, with a sudden decrease to 1.5 nm and an increase to 2.25 nm at ~32 ns. It was then observed to gradually increase until 3.25 nm at ~50 ns, after which it reached stability and continued to increase to ~3 nm until 100 ns. The D749N mutant complex was stable at 0.25–0.5 nm until 40 ns. At ~41 ns, there was an increase, with the peak reaching 1 nm, followed by a sudden decrease back to 0.5 nm, and the peak continued to be in the range of 0.5–1 nm until 100 ns. The R786Q mutant had the maximum fluctuation during the course of the simulation. At 10 ns, the R786Q mutant complex reached ~1.4 nm, further reaching 2.75 nm at 20 ns. From 20 to 50 ns, there was a gradual increase with many fluctuations, with the highest peak value of 3.3 nm at ~35 ns and the lowest value of 2.5 nm at 30 ns. After 50 ns, a further increase of 4.5 nm was observed at 100 ns. The RMSF analysis highlighted the flexibility and mobility of individual amino acid residues within the ITGB3 wild-type and mutant proteins and their interaction with calpain-1 (Figure 10b). In the wild-type ITGB3 (black)–Calpain-1 (red) complex, lower RMSF values were observed, indicating stable regions with limited flexibility. Since the wild-type complex exhibited controlled fluctuations, it suggests strong interactions and minimal disturbance to the protein structure. Conversely, the mutant ITGB3–Calpain-1 complex demonstrated pronounced peaks, indicating regions with increased flexibility and instability. The D749N (green) and R786Q (yellow) mutants exhibited significant residue fluctuations, especially in specific regions, indicating increased flexibility and potential destabilization of secondary structures. These peaks, which are particularly noticeable towards the terminal regions, suggest that mutations introduced additional mobility to the protein, which could adversely impact the stability and function of the mutant ITGB3–calpain-1 complex. The fluctuations in these specific regions could correlate with altered binding interactions, affecting the overall dynamics of the receptor–ligand complex. The radius of gyration (Rg) for ITGB3–calpain-1 was compared for the wild-type and mutant complexes (Figure 10c). The wild-type (black) complex, in the initial timestep, is in the range of 5.5–6 nm until 15 ns, followed by a gradual decrease to 4.5 nm at approximately 40–50 nm. From 50 to 100 ns, the wild-type was observed to be stable at ~4.5 nm throughout the simulation. The mutant D749N (red) complex was stable throughout the course of the simulation and was in the range of 4.25–4.45 nm, whereas the mutant R786Q (green) complex showed very high fluctuations. The R786Q mutant started at 7.25 nm, which decreased with increasing fluctuations in the initial time frame of 0–50 ns. Its highest peak was observed at 7.5 nm at ~5 ns, with a decrease to ~6.6 nm at 12 ns followed by a decrease to 5.6 nm at ~22 ns. There was a further decrease to 5 nm at ~35 ns, and then it remained within the same range of 5–5.5 nm from 35 to 50 ns. There was a gradual decrease to 4.5 nm from 70 to 80 nm, and from 80 ns, it varied between 5 and 4.7 nm till 100 ns. Overall, the mutant D749N complex was more compact and stable than the wild-type and mutant R786Q complexes were, whereas the mutant R786Q complex was the least stable among all. Hydrogen bond analysis provided critical insights into the strength and stability of interactions between ITGB3 receptors and calpain-1 (Figure 10d). The wild-type (black) ITGB3–Calpain complex consistently maintained a greater number of hydrogen bonds throughout the simulation, which is indicative of strong and stable binding interactions. This high hydrogen bond count reflects robust binding affinity and structural integrity, underscoring the stability of the wild-type complex. In contrast, the mutant complexes (D749N) (red) and (R786Q) (green) exhibited fewer hydrogen bonds, as depicted by the green and red lines, respectively, suggesting weakened or altered binding interactions due to the mutations. The reduction in hydrogen bonds highlights the disruptive effects of the mutations on the ITGB3–calpain-1 interaction, potentially impacting the biological functionality and stability of the complex. These findings suggest that the mutations compromised the binding affinity, leading to less stable interactions and altered dynamics within the receptor–ligand complex, which could have significant implications for ITGB3 functional activity and potential therapeutic targeting.

The binding free energy of the receptor–ligand complex was calculated using the MM/GBSA method for the last 20 ns of the simulation. In the case of wild-type ITGB1–calpain-2, the total binding free energy of the complex was −91.58 kcal/mol (Figure 11a), whereas in the mutant complex, it was −23.05 kcal/mol (Figure 11b). The overall binding free energy was considerably less negative in the mutant complex than in the wild-type, indicating a weaker binding affinity between the mutant ITGB1 and calpain-2, which shows that the mutation weakened the interaction of the mutant receptor–ligand complex. The binding free energy analysis of the ITGB7–calpain-2 complexes revealed that the wild-type complex has a moderate binding affinity with a total energy of −33.66 kcal/mol, driven by favorable van der Waals interactions (−143.81 kcal/mol) and solvation energy (−25.12 kcal/mol) but hindered by significant electrostatic repulsion (135.27 kcal/mol) (Figure 12a). In comparison, the mutant ITGB7–Calpain complex displayed a reduced binding affinity with a total energy of −43.25 kcal/mol, characterized by decreased van der Waals interactions (−97.22 kcal/mol) and less unfavorable electrostatic interactions (77.47 kcal/mol) (Figure 12b). These findings highlight that mutation weakened the hydrophobic interactions and slightly alleviated the electrostatic repulsion, resulting in a less stable complex. Binding free energy analysis of the ITGB3–calpain-1 complexes revealed that the wild-type complex has strong binding affinity, primarily driven by favorable van der Waals interactions and solvation energy, resulting in a total binding energy of −133.25 kcal/mol (Figure 13a). In contrast, the D749N mutation significantly weakened these interactions, leading to a much lower binding affinity with a total energy of −46.81 kcal/mol, highlighting the disruptive impact of the mutation on hydrophobic interactions (Figure 13b). The R786Q mutant complex exhibited moderate binding characteristics, with van der Waals interactions remaining relatively strong but offset by more unfavorable electrostatic interactions, yielding a total binding energy of −93.6 kcal/mol (Figure 13c). Among the mutations that were imposed on the wild-type receptor protein IGB3, R786Q showed better binding strength than D749N, but both were weaker than the wild-type interaction. The results indicate that these mutations were not favored over the wild-type in the context of their interaction with the calpain protein.

## 3. Discussion

The acquisition and accumulation of genetic mutations is a hallmark of all the cancer types. Studies have shown that each cancer type is associated with a signature mutational pattern (Alexandrov et al., 2020) [2]. Therefore, for the effective cancer management and to prolong the life expectancy of the cancer patients, it is essential to understand the mutational pattern specific for each cancer type to delineate the spatiotemporal dynamics of cancer pathophysiology.

A recent study reported the occurrence of a maximum number of missense mutations in and around the genetic region encoding the calpain cleavage site termed as the calpain cleavage-related mutations (CCRMs) in human calpain substrates (Liu et al., 2019) [30]. However, the impact of these mutations on the calpain–substrate interaction and the physiological role it plays have not yet been investigated. The proposed study investigated the impact of the calpain cleavage-related missense mutations on the calpain–substrate interaction and alterations in the folding pattern and binding affinity. Other similar studies have shown the impact of pathogenic point mutations on the sensitivity of substrates to proteases (Tompa et al., 2004) [31]. One of such studies reported the difference in the sensitivity of p53 mutants towards calpain, with some mutants depicting low sensitivity while other mutants being highly sensitive (Pariat et al., 1997) [32]. Some point mutations have been shown to impact the thermostability of enzymes such as lipase (Singh et al., 2016) [33]. Recently, several studies have addressed the impact of point mutations on the structure of proteins and their physiological interactions to evaluate how these mutations contribute to the disease pathogenesis. For instance, point mutations in the spike protein of the COVID-19 virus were shown to either strengthen or weaken its interaction with the ACE-2 receptor 9 (Rucker et al., 2023) [34]. In cancers, the oncogenic point mutations affect the binding kinetics of the protein–protein interaction network, causing dysfunction of the intrinsic apoptotic pathway (Zhao et al., 2015) [35]. In the case of Cutis Laxa, an inherited disorder of the connective tissue, a point mutation R119G in the enzyme Pyrroline-5-carboxylate reductase has been shown to weaken the flexibility of the enzyme near the catalytic pocket. This in turn impairs the binding of the cofactor nicotinamide adenine dinucleotide (NAD), leading to lowered affinity towards the same (Li et al., 2017) [36]. Five highly deleterious point mutations (R24C, Y180H, A205T, R210P, and R246C) identified using an integrated computational approach in the cyclin-dependent kinase-4 have been shown to affect its interaction with the ligand, flavopiridol (Nagasundaram et al., 2015) [37]. Moreover, two point mutations, T193I and R148H, in the human prion proteins, which are linked to the occurrence of familial Creutzfeldt–Jacob disease, impact the secondary structure, leading to the reduced conformational space (Borgohain et al., 2016) [4]. Thus, point mutations that are pathogenic or deleterious may impact the structural stability of a protein, further impairing the protein–protein interaction and downstream signaling pathway.

Integrins are cell surface receptors that mediate bidirectional signaling and play important roles in cellular activities such as cell proliferation, migration, angiogenesis, and apoptosis [30,38]. Conditional knockdown of ITGB3 reduces various phenotypes, such as cell migration, proliferation, and senescence, reducing angiogenesis and tumor growth (Kovacheva et al., 2021) [25]. Mutations in ITGB1 have been shown to convert benign skin tumors into malignant ones (Ferreira et al., 2009) [14]. The cytoplasmic domain of these receptors plays a crucial role in downstream signaling, and mutations in these domains have previously been shown to disrupt various cellular processes. One of the studies (Fitzpatrick et al., 2014) [39] reported the importance of the C-terminal domain of ITGB1 for interaction with the adapter protein kindlin-2. Similarly, (Pfaff et al., 1999) [40], stated that the cytosolic domain of ITGB3 is crucial for cell adhesion, spreading, endocytosis, and phagocytosis (Ylänne et al., 1995) [41]. A point mutation in ITGB6 was found to reduce its localization to the focal contacts (Huang et al., 1995) [21].

Calpains have been shown to cleave integrin-β subunits. Treatment of recombinant cytoplasmic domains of different integrin-β subunits with purified calpain resulted in the cleavage of these domains. Further analysis showed the clustering of these cleavage sites in the C-terminal half region and flanked by NPXY/NXXY (X referring to an amino acid) conserved motifs (Du et al., 1995; Pfaff et al., 1999) [12,40]. Mutations in these regions may also alter the recruitment and activation of adaptor and effector proteins, which otherwise mediate effective signal transduction (Qiu et al., 2019) [42].

The proposed study focused on the impact of CCRMs in the cytoplasmic domain of β-integrins and their interaction with calpains. The reported calpain cleavage sites in ITGB1, ITGB3, and ITGB7 are located near the carboxyl terminus, which belongs to the cytoplasmic domain of the molecule. The list of CCRMs in UCEC retrieved from TCGA revealed the occurrence of the majority of such mutations in and around the calpain cleavage sites of ITGB1 (T777M), ITGB3 (D749N and R786Q), and ITGB7 (R760H). The mutations investigated in the current study are close to these conserved motifs (NPXY/NXXY), highlighting the necessity to understand the alterations in the interaction between calpains and β-integrins and its subsequent impact on tumor invasion and metastasis. These mutations also need to be investigated given the fact that ITGB1 and ITGB7 are cancer-associated genes and are FDA-approved drug targets (Pang et al., 2023) [43].

Calpain dysregulation has been shown to promote the occurrence and spread of various cancer types (Shapovalov et al., 2022; Sharma et al., 2023; Shiraha et al., 2002) [44,45,46]. Various selective calpain inhibitors have been shown to circumvent cancer metastasis and even the occurrence of drug resistance (Carragher, 2006) [6]. A recent study (te Boekhorst et al., 2022) [47] highlighted the role of calpain-2 in the regulation of integrin–cytoskeletal interactions under normoxic and hypoxic conditions and its effect on cell migration during pathological conditions such as cancer. The cytoplasmic domain of the β-integrin subunit has been reported to have the utmost importance in cellular adhesion. One of these studies addressed the different substitutions in the β1 and β3 cytoplasmic domains expressed in the context of interleukin-2 receptor chimeras to address the impact of these mutations on different parameters, such as adhesion, cell attachment, and spreading. The study reported that some mutations have minimal impacts on these parameters, whereas other mutations have drastic impacts (Bodeau et al., 2001) [3]. Similarly, mutations in ITGB1, ITGB3, and ITGB7 can alter their interaction with calpain, which may affect the focal adhesion of cancer cells and therefore needs to be validated further via a multidisciplinary approach. The findings reported in the current study are derived from an extensive in silico-based approach; therefore, the data need further evaluation using an in vitro-based approach. A previous study reported the importance of the cytoplasmic domain of β3 in αIIbβ3 and the impact of mutation in its COOH terminus, leading to failure of cell spreading and adhesion (Ylänne et al., 1995) [41]. Given the physiological importance of integrins in normal and cancerous environments, their interaction with calpain leading to intracellular signal transduction and association with other molecules influencing cellular attachment to the extracellular matrix, and the role played by the carboxyl terminus in their cytoplasmic domain in interaction with calpain, any alterations in their structure due to mutations need to be comprehended.

## 4. Materials and Methods

### 4.1. Data Retrieval and Curation

A dataset of 88 human calpain substrates with their calpain cleavage sites was retrieved (Appendix A). A list of 21198 missense mutations identified in the uterine corpus endometrial carcinoma was retrieved from the TCGA-GDC database considering the following filters: pathogenic, deleterious, and probably damaging. Comparing these two datasets, a list of 48 calpain substrates with a total of 176 missense mutations in the calpain cleavage sites was constructed. The site of mutations was also analyzed to determine whether the missense mutations in each substrate were in or around the calpain cleavage site (Appendix A).

These 48 calpain protein substrates were further shortlisted on the basis of their involvement in activities related to tumor development or progression and whether the mutation is in the calpain cleavage site. Among the shortlisted substrates, 3 proteins, namely, ITGB1, ITGB7, and ITGB3, with missense mutations in calpain cleavage sites (Appendix A), were selected further to study alterations in folding patterns and protein–protein interactions (Table 7).

### 4.2. Pathogenicity Prediction

The pathogenic mutations reported in the calpain substrates ITGB1, ITGB3, and ITGB7 were T777M, R786Q and D749N, and R760H, respectively. All the mutations were further evaluated to predict their pathogenicity via in silico pathogenicity prediction tools such as Polyphen2, Fathmm, and Panther-PSEP. Although no pathogenicity prediction tool is 100% accurate, an average analysis can be performed to validate and verify the pathogenicity of the mutations via different platforms.

#### 4.2.1. PolyPhen-2

PolyPhen-2 prediction studies the impact of an amino acid on the structure and function of a protein via machine learning methods. The classification is reported as benign, possibly damaging, and probably damaging (Adzhubei et al., 2013) [1].

#### 4.2.2. Fathmm

The impact of a protein missense mutation on its function is predicted by combining sequence conservation with hidden Markov models (HMMs) (Rogers et al., 2018) [48]. The prediction threshold value is −0.75. Any value less than −0.75 predicts that the mutation is potentially associated with cancer.

#### 4.2.3. Panther-PSEP (Protein Analysis Through Evolutionary Relationships)

Nonsynonymous genetic variants that may play a role in human diseases can be predicted via position-specific evolutionary preservation (Tang and Thomas, 2016) [49].

The protein sequences of wild-type ITGB1, ITGB3, and ITGB7 were retrieved from the UniProt Knowledgebase with UniProt IDs P05556, P05106, and P26010, respectively. The mutation in ITGB1 was introduced by mutating threonine at position 777 to methionine (T777M), and the mutation in ITGB3 was introduced by mutating arginine to glutamine at the 786th position (R786Q) or aspartic acid to asparagine (D749N) at the 749th position, whereas in ITGB7, arginine at position 760 was mutated to histidine (R760H) in PyMOL (Schrodinger, L. (2010) The PyMOL Molecular Graphics System, Version 1.3r1., n.d.) [29].

### 4.3. Calpain Cleavage Prediction

DeepCalpain (Liu et al., 2019) [30] was used to identify and compare the calpain cleavage sites of the wild-type and mutant substrate proteins. DeepCalpain is a web server based on a deep neural network and the particle swarm optimization algorithm, which predicts the potential calpain cleavage sites in a given protein sequence. The ITGB1, ITGB3, and ITGB7 wild-type and mutant sequences were submitted to DeepCalpain, and the output was analyzed. Finally, the results obtained for the wild-type and mutant sequences were compared with respect to the exact cleavage position and number of cleavage sites.

### 4.4. Structure Retrieval

The beta integrin subunits ITGB1 and ITGB7 are substrates of calpain-2, whereas ITGB3 is a substrate of calpain-1. Therefore, the structures of the human calpain-1, calpain-2, and beta integrin substrates were retrieved from the PDB/AlphaFold (Jumper et al., 2021) [24] database for docking as follows: (1) PDB ID: 3DF0, a calcium-dependent complex between calpain-2 and calpastatin from *Rattus norvegicus,* was retrieved. The protein sequence of rat calpain-2 was 93.7% identical and 97.4% similar to that of the human calpain-2 (Appendix A); therefore, it was modeled to that of human calpain-2 by mutating the corresponding amino acid. PDB ID: 3DF0 comprises Chain-A (catalytic subunit of m-calpain), Chain-B (regulatory subunit of m-calpain), Chain-C (calpastatin), and calcium ligands. Chain-B and Chain-C were deleted from the PDB ID: 3DF0, and Chain-A was modified according to the sequence of human calpain-2 to derive the catalytic domain of calcium-dependent human calpain-2. (2) AlphaFold ID: AF-P07384, the structure of the catalytic subunit of calpain-1 was retrieved from the AlphaFold database (Jumper et al., 2021) [19], and subsequently, calcium molecules were docked against the same structure with reference to calpain-2. (3) The structures of calpain substrates, ITGB1 (AF-P05556-F1), ITGB3 (AF-P05106-F1), and ITGB7 (AF-P26010-F1), were retrieved from AlphaFold (Jumper et al., 2021) [24], and mutations in the substrate proteins were introduced into PyMOL (Schrodinger, L. (2010) The PyMOL Molecular Graphics System, Version 1.3r1., n.d.).

### 4.5. Modeling and Simulation

The final structures of calpain-1 and calpain-2 were modeled and simulated for better stability prior to docking (Appendix A). Although the full-length structures of ITGB1, ITGB3, and ITGB7 retrieved from AlphaFold (Jumper et al., 2021) [24] had average predicted local distance difference test (pLDDT) scores of 85.88, 87.31, and 82.67, respectively, the structures were further modeled and simulated to obtain stable confirmation (Appendix A). Modeling and simulation of all the structures were performed with Gromacs 2021.2 software for 100 ns before docking. For MD simulation, each protein was solvated in a cubic water box with periodic boundary conditions, and ions were added to neutralize the system. The CHARMM27 force field was used for the protein, and the TIP3P water model for the solvent. The system was energy minimized to remove any bad contacts or steric clashes. The system was equilibrated in two phases: NVT (constant number, volume, and temperature) for 1 ns and NPT (constant number, pressure, and temperature) for 1 ns ps. A 100 ns MD simulation was performed for each protein at 310 K using the NPT ensemble. The conformational stability of the wild-type and mutant substrates was evaluated by the RMSD, RMSF, and radius of gyration (Rg) values.

### 4.6. Molecular Docking and MD Simulation of Docked Complexes

Protein–protein docking of shortlisted substrates to specific calpains was performed via HDock (Yan et al., 2017) [50] to study the interface between calpain and wild-type/mutant substrate proteins. ITGB1 and ITGB7 wild-type, along with their mutants, T777M and R760H, respectively, were docked with calpain-2. Similarly, wild-type ITGB3 and its mutants R786Q and D749N were docked with calpain-1. Targeted docking was performed on a core protease domain of calpain-1 and calpain-2 and substrate regions with experimentally known calpain cleavage sites, followed by molecular dynamic simulation of the docked complexes. The results obtained from molecular docking were confirmed by MD simulations of docked complexes using Gromacs 2021.2 software for 100 ns. Further post-MD analysis was carried out. The post MD-analysis was performed on the visual platform called ‘Analogue’, developed by Growdea Technologies [51,52] (https://growdeatech.com/Analogue/, accessed on 23 August 2024).

## Figures and Tables

**Figure 1 ijms-26-04246-f001:**
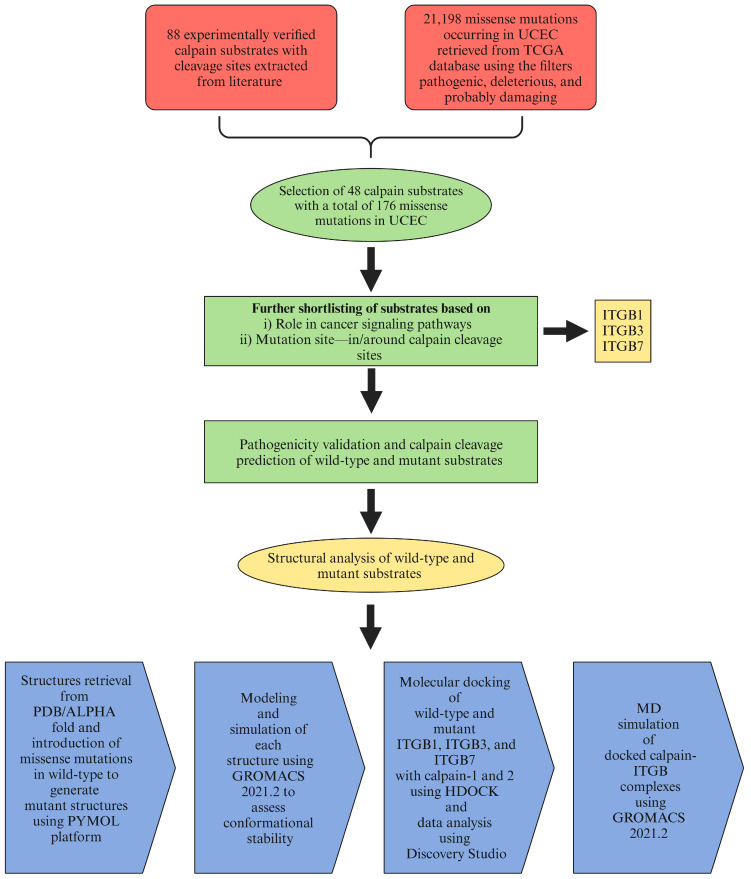
Schematic representation of the workflow.

**Figure 2 ijms-26-04246-f002:**
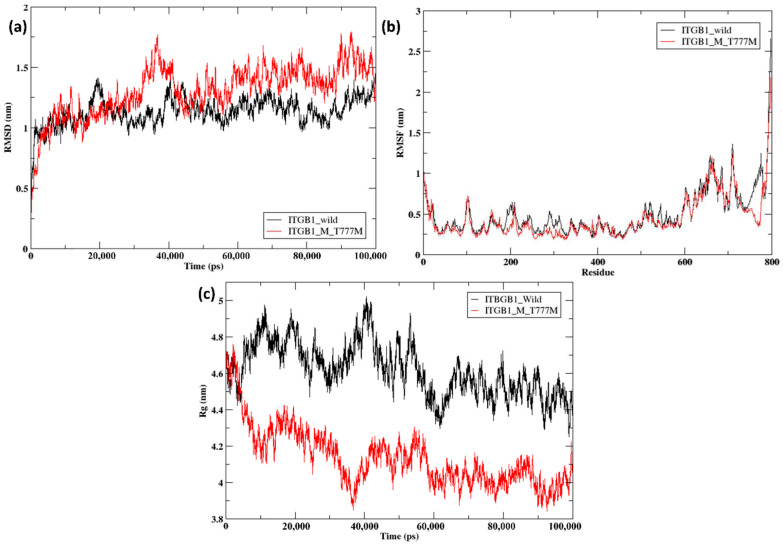
(**a**) RMSD of ITGB1 wild-type (black) and mutant (T777M) (red); (**b**) RMSF of ITGB1 wild-type (black) and mutant (T777M) (red); (**c**) Rg of ITGB1 wild-type (black) and mutant (T777M) (red).

**Figure 3 ijms-26-04246-f003:**
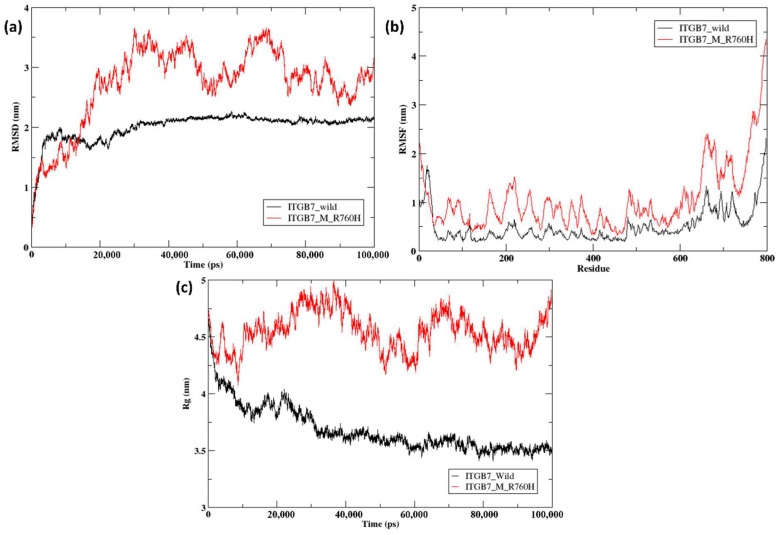
(**a**) RMSD of wild-type (black) and mutant ITGB7 R760H (red); (**b**) RMSF of wild-type (black) and mutant ITGB7 R760H (red); (**c**) Rg of wild-type (black) and mutant ITGB7 R760H (red).

**Figure 4 ijms-26-04246-f004:**
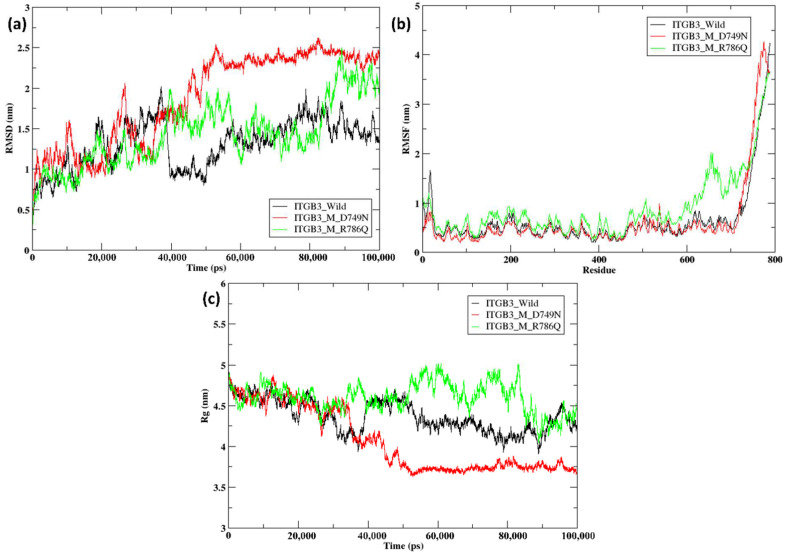
(**a**) RMSD of wild-type (black) and mutant ITGB3 (D749N) (red) and (R786Q) (green); (**b**) RMSF of wild-type (black) and mutant ITGB3 (D749N) (red) and (R786Q) (green); (**c**) Rg of wild-type (black) and mutant ITGB3 (D749N) (red) and (R786Q) (green).

**Figure 5 ijms-26-04246-f005:**
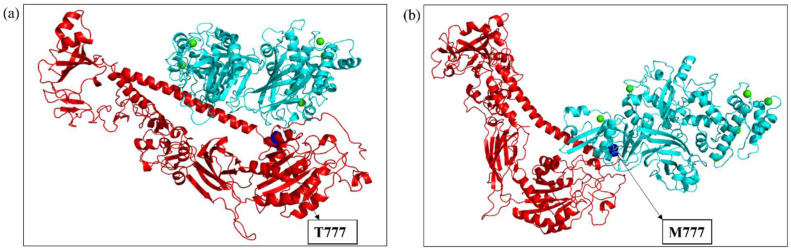
(**a**) Docked complex of wild-type ITGB1 (red) and calpain-2 (cyan); (**b**) docked complex of the ITGB1 mutant (T777M) (red) and calpain-2 (cyan). Blue spheres depict the mutation and position. Green spheres depict the calcium bound to calpain-2.

**Figure 6 ijms-26-04246-f006:**
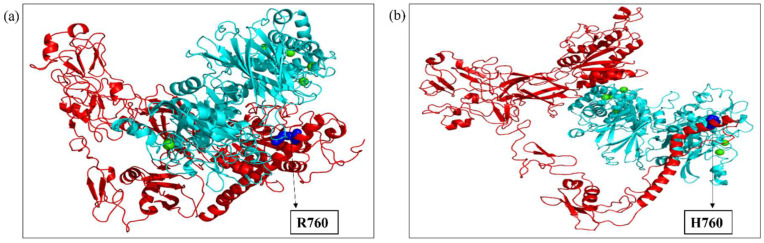
(**a**) Docked complex of wild-type ITGB7 (red) and calpain-2 (cyan); (**b**) docked complex of the ITGB7 mutant (R760H) (red) and calpain-2 (cyan). Blue spheres depict the mutation and position. Green spheres depict the calcium bound to calpain-2.

**Figure 7 ijms-26-04246-f007:**
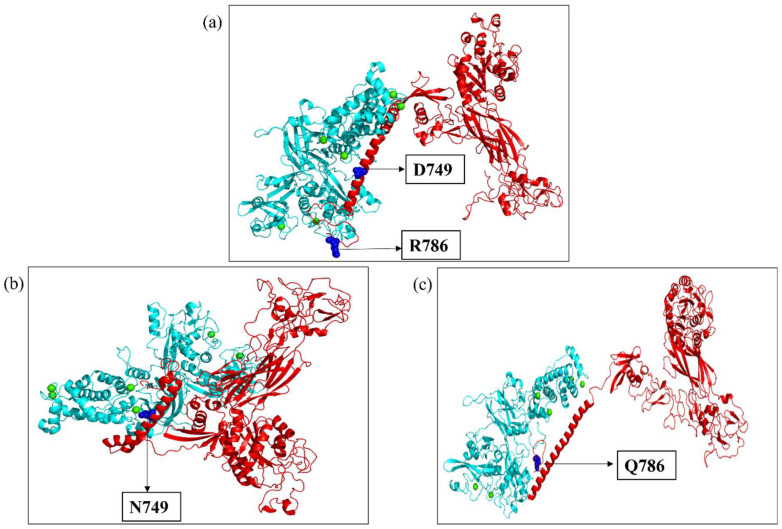
(**a**) Docked complex of wild-type ITGB3 (red) and calpain-1 (cyan); (**b**) docked complex of the ITGB3 mutant (D749N) (red) and calpain-1 (cyan); (**c**) docked complex of the ITGB3 mutant (R786Q) (red) and calpain-1 (cyan). Blue spheres depict the mutation and position. Green spheres depict the calcium bound to calpain-1.

**Figure 8 ijms-26-04246-f008:**
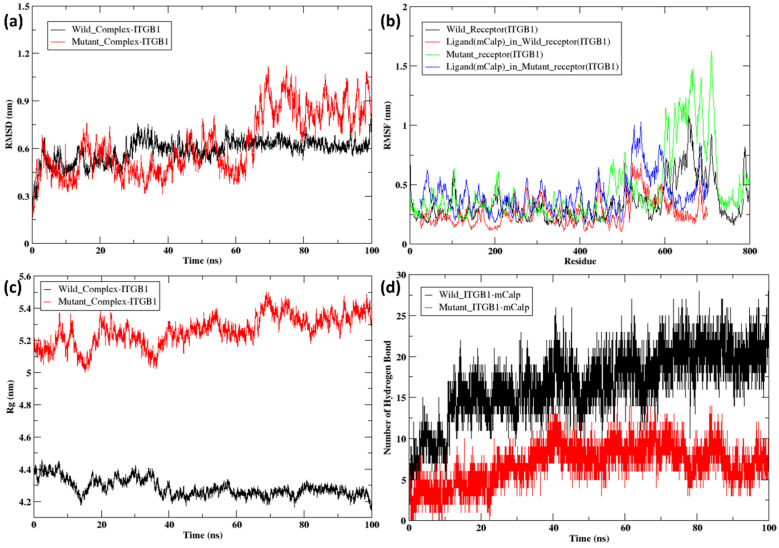
(**a**) RMSD of the docked complex of wild-type (black) and mutant ITGB1 (T777M) (red); (**b**) RMSF of the docked complex of wild-type (black) with calpain-2/m-calpain (red) and mutant ITGB1 (T777M) (green) with calpain-2/m-calpain (blue); (**c**) Rg of the docked complex of wild-type (black) and mutant ITGB1 (T777M) (red); (**d**) hydrogen bond of the docked complex of wild-type (black) and mutant ITGB1 (T777M) (red).

**Figure 9 ijms-26-04246-f009:**
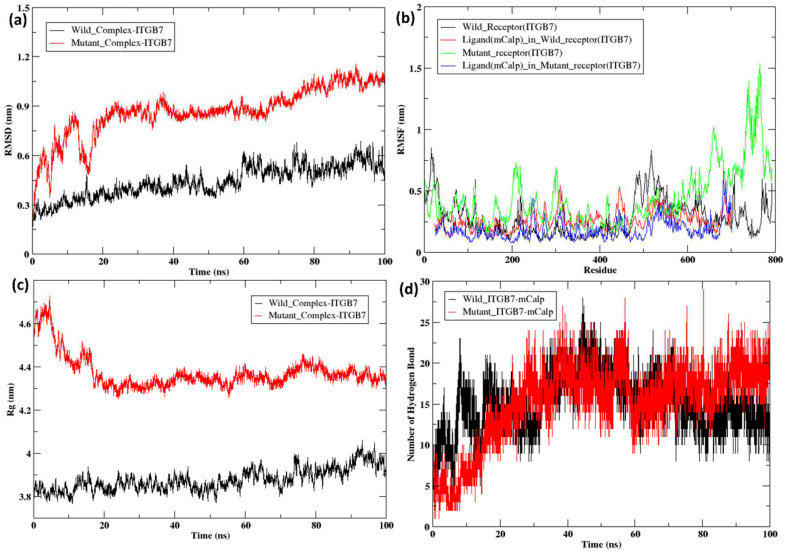
(**a**) RMSD of the docked complex of wild-type (black) and mutant ITGB7 (R760H) (red); (**b**) RMSF of the docked complex of wild-type (black) with calpain-2/m-calpain (red) and mutant ITGB7 (R760H) (green) with calpain-2/m-calpain (blue); (**c**) Rg of the docked complex of wild-type (black) and mutant ITGB7 (R760H) (red); (**d**) hydrogen bond of the docked complex of wild-type (black) and mutant ITGB7 (R760H) (red).

**Figure 10 ijms-26-04246-f010:**
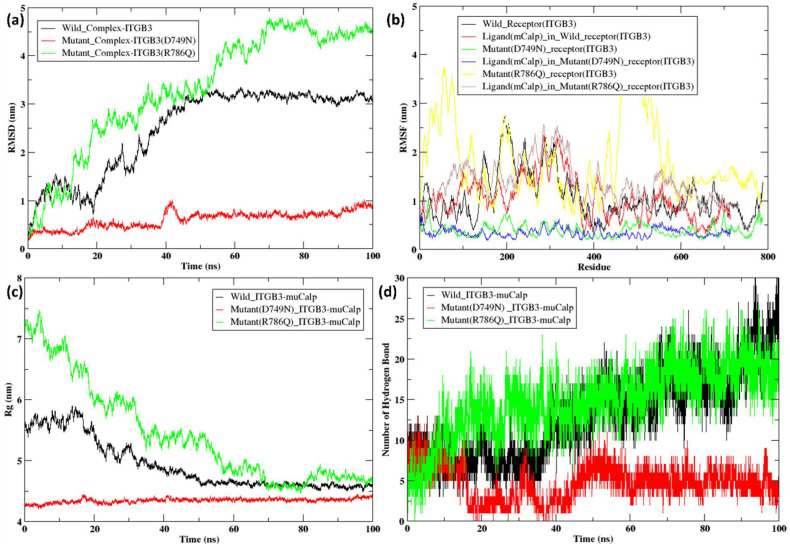
(**a**) RMSD of the docked complex of wild-type (black) and mutant ITGB3 (D749N) (red) (R786Q) (green); (**b**) RMSF of the docked complex of wild-type (black) with calpain-1/mu-calpain (red) and mutant ITGB3 (D749N) (green) with calpain-2/m-calpain (blue), (R786Q) (yellow) with calpain-2/m-calpain (brown); (**c**) Rg of the docked complex of wild-type (black) and mutant ITGB3 (D749N) (red) (R786Q) (green); (**d**) hydrogen bond of the docked complex of wild-type (black) and mutant ITGB3 (D749N) (red) (R786Q) (green).

**Figure 11 ijms-26-04246-f011:**
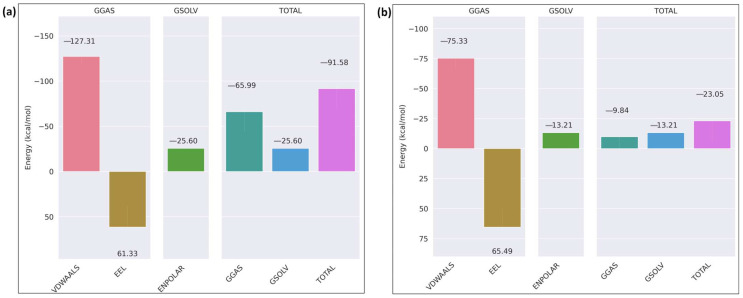
MMGBSA binding free energy of ITGB1: (**a**) wild-type; (**b**) mutant (T777M).

**Figure 12 ijms-26-04246-f012:**
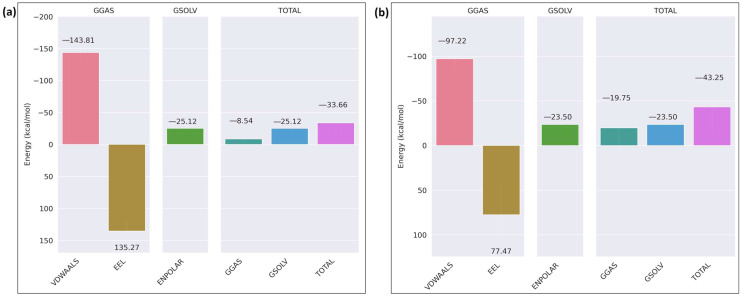
MMGBSA binding free energy of ITGB7: (**a**) wild-type; (**b**) mutant (R760H).

**Figure 13 ijms-26-04246-f013:**
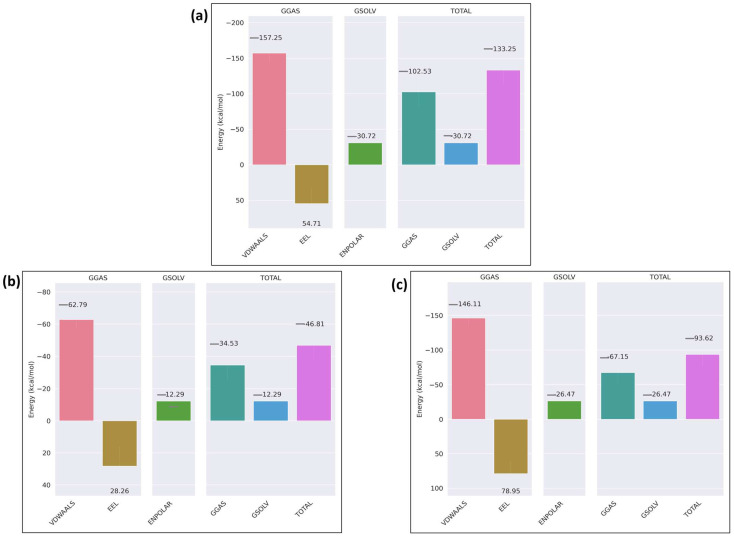
MMGBSA binding free energy of ITGB3: (**a**) wild-type; (**b**) mutant_D749N; (**c**) mutant_R786Q.

**Table 1 ijms-26-04246-t001:** Pathogenicity verification of selected mutations.

Substrates	Mutations	Polyphen2 (0–1)	Fathmm (−0.75)	Panther
ITGB1	T777M	Most likely damaging (0.998)	Cancer (−1.03)	Damaging (0.316)
ITGB7	R760H	Most likely damaging (1.00)	Cancer (−1.31)	Damaging (0.280)
ITGB3	D749N	Most likely damaging (1.00)	Cancer (−0.98)	Most likely damaging (0.85)
	R786Q	Most likely damaging (1.00)	Cancer (−0.98)	Most likely damaging (0.57)

**Table 2 ijms-26-04246-t002:** Calpain cleavage site prediction by DeepCalpain.

Substrates	Site of Mutation	Total Number of Calpain Cleavage Sites Predicted	Calpain Cleavage Site Variation Around Mutation Site	
			Wild-Type	Mutant
ITGB1	T777M	Wild-type: 05 Mutant: 03	778–779; 788–789: cleavage predicted	778–779; 788–789: cleavage not predicted
ITGB7	R760H	Wild-type: 06 Mutant: 05	769–770: cleavage predicted	769–770: cleavage not predicted
ITGB3	D749N	Wild-type: 12 Mutant: 12	No variation predicted	No variation predicted
	R786Q	Wild-type: 12 Mutant: 10	772–773; 779–780: cleavage predicted	772–773; 779–780: cleavage not predicted

**Table 3 ijms-26-04246-t003:** Variations in calpain cleavage sites, binding affinity, RMSD, hydrogen bonds, molecular interactions, and radius of gyrations between wild-type and mutant ITGB1.

Parameters	Wild-Type ITGB1	Mutant ITGB1 (T777M)
DeepCalp cleavage sites	5	3
Docking score	−222.61	−206.58
Confidence score	0.8103	0.7561
RMSD	1.227Å	
H-bonds	8	4
Hydrophobic interactions	6	7
Electrostatic interactions	9	2
Average radius of gyrtaion	4.72 nm	4.19 nm

**Table 4 ijms-26-04246-t004:** Variations in calpain cleavage sites, binding affinity, RMSD, hydrogen bonds, molecular interactions, and radius of gyrations between wild-type and mutant ITGB7.

Parameters	Wild-Type ITGB7	Mutant ITGB7 (R760H)
DeepCalp cleavage sites	6	5
Docking score	−222.31	−222.50
Confidence score	0.8094	0.8100
RMSD	1.081Å	
H-bonds	10	6
Hydrophobic interactions	9	3
Electrostatic interactions	6	5
Average radius of gyrtaion	3.72 nm	4.60 nm

**Table 5 ijms-26-04246-t005:** Variations in calpain cleavage sites, binding affinity, RMSD, hydrogen bonds, molecular interactions, and radius of gyrations between wild-type and mutant ITGB3.

Parameters	Wild-Type ITGB3	Mutant ITGB3 D749N	Mutant ITGB3 R786Q
DeepCalp cleavage sites	12	12	10
Docking score	−283.58	−234.95	−224.14
Confidence score	0.935	0.845	0.815
RMSD		1.013	1.071
H-bonds	12	8	6
Hydrophobic interactions	9	6	8
Electrostatic interactions	5	3	1
Average radius of gyrtaion	4.38 nm	4.07 nm	4.56 nm

**Table 6 ijms-26-04246-t006:** (**a**) Interface residues of the ITGB1–Calpain-2 complex; (**b**) interface residues of the ITGB7–Calpain-2 complex; (**c**) interface residues of the ITGB3–Calpain-1 complex.

(a)		(b)		(c)		
Wild-Type	Mutant T777M	Wild-Type	Mutant R760H	Wild-Type	Mutant D749N	Mutant R786Q
LEU753	-	ARG747	-	LYS742	LYS742	-
LEU754	-	SER749	SER749	LEU743	-	-
MET755	-	VAL750	VAL750	LEU744	-	-
ILE756	-	GLU751	GLU751	-	ILE745	-
ILE757	-	TYR753	TYR753	THR746	THR746	-
HIS758	HIS758	ASP754	-	ILE747	-	-
ASP759	-	-	ARG756	-	HIS748	-
ARG761	ARG761	GLU757	GLU757	ASP749	ASN749	-
GLU762	GLU762	-	HIS760	ARG750	-	-
ALA764	-	GLU764	-	LYS751	-	-
LYS765	LYS765	GLN767	-	-	GLU752	-
LYS768	LYS768	LEU768	-	PHE753	PHE753	PHE753
GLU769	GLU769	-	TRP770	-	PHE756	PHE756
LYS770	-	LYS771	LYS771	GLU757	-	GLU757
MET771	-	-	ASN775	GLU758	-	GLU758
ASN772	ASN772	-	TYR778	ARG760	-	ARG760
ALA773	-	LYS779	LYS779	ALA761	-	ALA761
LYS774	LYS774	SER780	SER780	ARG762	-	-
TRP775	TRP775	ALA781	ALA781	ALA763	-	-
ASP776	ASP776	-	-	-	-	LYS764
THR777	MET777	-	-	-	-	TRP765
-	GLY778			THR767	-	THR767
GLU779	GLU779			ALA768	-	ALA768
ASN780	ASN780			ASN769	-	-
-	PRO781			ASN770	-	-
TYYR783	-			PRO771	-	-
-	LYS784			LEU772	-	-
-	ALA786			-	-	TYR773
-	VAL787			LYS774	-	LYS774
-	THR788			GLU775	-	-
-	THR789			ALA776	-	-
				THR777	THR777	THR777
				SER778	SER778	SER778
				-	-	THR779
				-	PHE780	PHE780
				-	THR781	THR781
				-	ASN782	ASN782
				-	ILE783	-
				-	THR784	-
				TYR785	TYR785	TYR785
				-	ARG786	GLN786
				-	GLY787	GLY787
				THR788	THR788	THR788

**Table 7 ijms-26-04246-t007:** Shortlisted calpain substrates with missense mutations in UCEC.

Sl. No	Protein Substrate	Selected Mutation	Calpain Cleavage Sites	Calpain Type
1.	ITGB1	T777M	771 (MN), 777 (TG), 778 (GE), 789 (TV), 791 (VN)	Calpain-2 (m-calpain)
2.	ITGB7	R760H	746 (YR), 760 (RF), 765 (QQ), 766 (QQ), 769 (NW), 770 (WK), 773 (DS), 774 (SN), 778 (YK), 784 (TT), 785 (TI)	Calpain-2 (m-calpain)
3.	ITGB3	D749N R786Q	761 (AR), 767 (TA), 773 (YK), 780 (FT), 785 (YR)	Calpain-1 (µ-calpain)

## Data Availability

The authors state that all the associated data are available in the manuscript.

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
