# Peer review of "Repercussions of the Calpain Cleavage-Related Missense Mutations in the Cytosolic Domains of Human Integrin-β Subunits on the Calpain–Integrin Signaling Axis"

_ijms, 2025, doi:10.3390/ijms26094246_

Round 1
Reviewer 1 Report
Comments and Suggestions for Authors
Comment 1: It is suggested that the list of authors be unified. In some cases, there is a full name and surname; in other cases, initials are used.
Comment 2: It is suggested that an abbreviation list can be added. It will make following the paper much easier.
Comment 3: It is suggested that the title of the figure be added after the abstract if it should be a part of the manuscript. Reference to this figure also should be placed in the text. Additionally, the font used in the figure is too small, and it isn't easy to analyze the figure.
Comment 4: It is suggested that fonts be unified in manuscripts (for example, in the titles of the figure where mutation is always written using larger font than the rest of the title).
Comment 5: It is suggested that color code information be added in the captions of Figures 1-3. It is mentioned in the manuscript, but adding it to the figure title would also be useful.
Comment 6: The authors should add some more details to the description of figures prepared using PyMol, for example, which protein is colored green and which cyan, what is marked as green dots. It would also be worth improving the resolution of those figures, marking the position of the mutation by showing this residue as spheres in different color. To compare presented complexes, it would be advantageous to show one of the proteins in the same position on both figures (a and b).
Comment 7: It is suggested that more descriptions be added to the titles of Tables 3-5.
Comment 8: It is suggested that Tables 6-8 could be more readable, for example, by placing interactions that are present in both complexes in the same line of the table, and when the interaction occurs in one of the complexes, leave the line empty for the complex where interaction is not present.
Comment 9: It is suggested that color code information be added in the captions of Figures 7-9. It is not mentioned in the manuscript, which color is used for wild-type protein and which for mutant.
Comment 10: In lines 69-70, the authors claim that “Human integrins are transmembrane heterodimeric glycoproteins crucial for cell–69 cell and cell–matrix interactions”. However, the role of glycosylation is not discussed in the manuscript.
Comment 11: The authors should add information about sequence differences between ITGB1, ITGB3 and ITGB7.
Comment 12: In line 721 the author wrote” (2) PDB ID: P07384” but this is not a PDB code.
Comment 13: For the structure of calpain-1, the structure of rat protein was used. Do the authors compare modified rat structure with human structure retrieved from AlphaFold??
Comment 14: The title of the manuscript “Repercussions of the Calpain Cleavage Related Missense Mutations in the Cytosolic domains of Human integrin-β subunits 2 on the Progression of Uterine Corpus Endometrial Carcinoma” suggest that the authors will discuss the impact on uterine corpus endometrial carcinoma but it is missing in the manuscript.
Comment 15: Some names are suggested to be unified in manuscripts. For example:
- “integrin-β subunit” (line 79), “Integrin-β subunit” (line 82) and “integrinβ-subunit” (line 109).
- AlphaFold (line 139), Alphafold (line 713)
Comment 16: It is suggested that the manuscript be improved to avoid editorial mistakes.
- not to use “&” sign in the manuscript;
- unified mutant descriptions in the manuscript (use three-letter code or one-letter for amino acid names in the whole manuscript)
- add references to the programs used not only in the “Material and methods” section but whenever they are in the text.
Comments on the Quality of English LanguageThe English could be improved to more clearly express the research
Author Response
Rebuttal for Reviewer-1
Thank you very much for taking the time to review this manuscript. Please find the detailed responses below and the corresponding revisions/corrections have been highlighted in yellow in the revised manuscript.
Comment 1: It is suggested that the list of authors be unified. In some cases, there is a full name and surname; in other cases, initials are used.
Rebuttal: The author names have been mentioned as observed in their other publications. However, as per the suggestion, we have changed it to a uniform style of First Name, followed by Middle Name and Last Name. For some authors, the middle name is provided in the form of initials to maintain uniformity in the way their names appeared in the previous publications.
Comment 2: It is suggested that an abbreviation list can be added. It will make following the paper much easier.
Rebuttal: Thank you for the valuable suggestion. Accordingly, a list of abbreviations has been added to the manuscript after the keywords (Lines 48-54).
Comment 3: It is suggested that the title of the figure be added after the abstract if it should be a part of the manuscript. Reference to this figure also should be placed in the text. Additionally, the font used in the figure is too small, and it isn't easy to analyze the figure.
Rebuttal: As per the reviewer’s suggestion, the figure added after the abstract has now been included in the manuscript as Figure 1. Furthermore, it has been modified for better clarity and has been provided with a title in the revised manuscript. The reference for the same has been placed in the text (Line 107).
Comment 4: It is suggested that fonts be unified in manuscripts (for example, in the titles of the figure where mutation is always written using larger font than the rest of the title).
Rebuttal: Thank you for pointing out the difference. As per the reviewer’s suggestion, the font for all mutations has been unified throughout the revised manuscript and is highlighted in yellow.
Comment 5: It is suggested that color code information be added in the captions of Figures 1-3. It is mentioned in the manuscript, but adding it to the figure title would also be useful.
Rebuttal: As per the reviewer’s suggestion, the color code information has been added to the captions of Figures 2-4 (Line 210-211; Line 214-215; Line 219-220) in the updated manuscript and is highlighted in yellow.
Please Note: Due to the inclusion of Figure 1 in the revised manuscript, Figure 1, Figure 2 and Figure 3 from the original manuscript have been changed to Figure 2, Figure 3, and Figure 4 respectively in the updated manuscript.
Comment 6: The authors should add some more details to the description of figures prepared using PyMol, for example, which protein is colored green and which cyan, is marked as green dots. It would also be worth improving the resolution of those figures, marking the position of the mutation by showing this residue as spheres in different colors. To compare presented complexes, it would be advantageous to show one of the proteins in the same position on both figures (a and b).
Rebuttal: Thank you for the valuable suggestion. Accordingly, more details have been added to the descriptions of the Figures. 5, 6, and 7) as provided below. Moreover, these figures prepared using pymol have been modified for better clarity and resolution. The mutations are shown as blue spheres and are labeled clearly. Furthermore, as suggested, one of the proteins in the docked complexes of the substrate (ITGB) and calpain, i.e., wild-type and mutant ITGB1, ITGB3, and ITGB7 are in a similar position in (a) and (b) of all figures for better comparison.
Descriptions of Figures 5, 6, and 7
Figure 5: (a) Docked complex of wild-type ITGB1 (red) and calpain-2 (cyan); (b) Docked complex of the ITGB1 mutant (T777M) (red) and calpain-2 (cyan). Blue spheres depict the mutation and position. Green spheres depict the calcium bound to calpain-2 (Line 286-288)
Figure 6: (a) Docked complex of wild-type ITGB7 (red) and calpain-2 (cyan); (b) docked complex of the ITGB7 mutant (R760H) (red) and calpain-2 (cyan). Blue spheres depict the mutation and position. Green spheres depict the calcium bound to calpain-2 (Line 297-299)
Figure 7: (a) Docked complex of wild-type ITGB3 (red) and calpain-1 (cyan); (b) docked complex of the ITGB3 mutant (D749N) (red) and calpain-1 (cyan); (c) docked complex of the ITGB3 mutant (R786Q) (red) and calpain-1 (cyan). Blue spheres depict the mutation and position. Green spheres depict the calcium bound to calpain-1 (Line 309-311)
Comment 7: It is suggested that more descriptions be added to the titles of Tables 3-5.
Rebuttal: As per the reviewer’s suggestion, more description has been added to the titles of Tables 3 (Line 292-293), Table 4 (Line 303-304), and Table 5 (Line 316-318) as follows:
Variations in calpain cleavage sites, binding affinity, RMSD, Hydrogen bonds, molecular interactions, and radius of gyrations between wild-type and mutant ITGB1, Table 3 (Line 292-293).
Variations in calpain cleavage sites, binding affinity, RMSD, Hydrogen bonds, molecular interactions, and radius of gyrations between wild-type and mutant ITGB7, Table 4 (Line 303-304).
Variations in calpain cleavage sites, binding affinity, RMSD, Hydrogen bonds, molecular interactions, and radius of gyrations between wild-type and mutants of ITGB3 Table 5 (Line 316-318).
Comment 8: It is suggested that Tables 6-8 could be more readable, for example, by placing interactions that are present in both complexes in the same line of the table, and when the interaction occurs in one of the complexes, leave the line empty for the complex where interaction is not present.
Rebuttal: We thank the reviewer for their valuable suggestion. As per the suggestion, the tables showing the interface residues of ITGB1-Calpain-2 complex, Table 6a (Line 355), ITGB7-Calpain-2 complex Table 6b (Line 385), and ITGB3-Calpain-1 complex Table 6c (Line 418) have been depicted by placing common interacting residues in the same line and leaving the line empty where there is no interaction in the respective complex.
Comment 9: It is suggested that color code information be added in the captions of Figures 7-9. It is not mentioned in the manuscript, which color is used for wild-type protein and which for mutant.
Rebuttal: As per the reviewer’s suggestion, the color code information for Figure 8 (Line 461-464), Figure 9 (Line 513-516) and Figure 10 (Line 575-579) has been added and the same has been reflected in the text as well (Line 428-458; Line 466-508; Line 518-572). The additions have also been highlighted in yellow.
Comment 10: In lines 69-70, the authors claim that “Human integrins are trans-membrane heterodimeric glycoproteins crucial for cell-cell and cell-matrix interactions”. However, the role of glycosylation is not discussed in the manuscript.
Rebuttal: As per the reviewer’s suggestion, the role of glycosylation in the beta subunit of integrins has been discussed in the manuscript (Lines 82-86) and highlighted in yellow.
Glycosylation of integrin beta subunits has been shown to play a key role in maintaining active conformation required for ligand binding and subsequent downstream signaling pathways. The degree and type of glycosylation of integrin beta subunits have been shown to change during cancer progression.
Comment 11: The authors should add information about sequence differences between ITGB1, ITGB3, and ITGB7.
Rebuttal: As per the reviewer’s suggestion, the sequences of wild-type and mutants of ITGB1, ITGB7, and ITGB3 have been provided as supplementary information in Supplementary Document S3. The exact position of the mutation has been colored in red.
Comment 12: In line 721 the author wrote” (2) PDB ID: P07384” but this is not a PDB code.
Rebuttal: Thank you for the valuable suggestion and correction. The given structure ID for the catalytic subunit of calpain-1 has been rectified to AlphaFold ID AF-P07384 (Line 790).
Comment 13: For the structure of calpain-1, the structure of rat protein was used. Do the authors compare modified rat structure with human structure retrieved from AlphaFold??
Rebuttal: Calpain-1/mu-calpain structure was retrieved from AlphaFold AF-P07384.
Since the structure of human calpain-2/m-calpain is not available, the following modifications were made. PDB ID: 3DF0, a calcium-dependent complex between calpain-2 and calpastatin from Rattus norvegicus, was retrieved from PDB. Chain B (Calpain small subunit 1) and Chain C (Calpastatin) were deleted. Chain A was modified as per the human calpain-2/m-calpain sequence as the identity is 93.7% and similarity is 97.4%. The information for the same has been provided to supplementary data (Supplementary Document S4) and the same has been incorporated in the text (Line 782-790).
Comment 14: The title of the manuscript “Repercussions of the Calpain Cleavage Related Missense Mutations in the Cytosolic domains of Human integrin-β Subunit 2 on the Progression of Uterine Corpus Endometrial Carcinoma” suggests that the authors will discuss the impact of uterine corpus endometrial carcinoma but it is missing in the manuscript.
Rebuttal: Thank you for the suggestion. Uterine Corpus Endometrial Carcinoma was incorporated in the title because the missense mutations observed in UCEC were retrieved and focused upon throughout the study. However, the impact of these mutations on the progression of UCEC requires wet lab experiments which are in progress.
Therefore, after considering the suggestion, the title of the manuscript has been modified to Repercussions of the Calpain Cleavage Related Missense Mutations in the Cytosolic domains of Human integrin-β subunits on the calpain-integrin signaling axis (Line 1-3).
Comment 15: Some names are suggested to be unified in manuscripts. For example:- “integrin-β subunit” (Line 79), “Integrin-β subunit” (Line 82), and “integrinβ-subunit” (Line 109).
- AlphaFold (Line 139), Alphafold (Line 713)
Rebuttal: Thank you for the suggestion. The differences in the names have been rectified and are made uniform.
integrin-β subunit (Line 2, 20, 21, 95,98, 277, 672, 673)
AlphaFold (Line 144, 781, 790, 791, 795, 802)
Comment 16: It is suggested that the manuscript be improved to avoid editorial mistakes.
- not to use “&” sign in the manuscript;
- unified mutant descriptions in the manuscript (use three-letter code or one-letter for amino acid names in the whole manuscript)
- add references to the programs used not only in the “Material and methods” section but whenever they are in the text.
Rebuttal: Thank you for the suggestion.
- As per the suggestion, the “&” sign has been replaced with ‘and’ throughout the revised manuscript. Moreover, utmost attention has been provided to avoid editorial errors.
- All the mutant descriptions have been unified as the one-letter symbol for the amino acids.
- References to the programs have been added to the material and methods section (Lines 753, 756, 761, 768, 771, 781, 795-796) and have been highlighted in yellow.
Reviewer 2 Report
Comments and Suggestions for Authors
This manuscript by Kizhakethil and Kamble et al. provides insights into the effect of mutation in the calpain cleavage of the calpain substrates on the calpain-substrate protein interactions. The authors have primarily picked up three calpain substrates, namely Integrin-β1 (ITGB-1), ITGB-1, and ITGB-1, to demonstrate the effect of the calpain cleavage site mutation in protein-protein interactions by employing molecular docking and molecular dynamics simulations experiments. This is an interesting study and may be beneficial in developing future calpain-integrin interaction inhibitors for UCEC. I would like to recommend the inclusion of this manuscript pending the following revisions.
General Comments
- Grammar and Spell Check: The manuscript should undergo a thorough grammar and spell check to enhance its readability.
Methodological Improvements
- Molecular Dynamics (MD) Simulations:
- MD simulations were performed for 100 ns, which may not provide adequate conformational sampling for systems such as ITGB1, ITGB3, and ITGB7, which are crucial to this study. I strongly recommend extending the simulations to 300–400 ns for wild-type and mutant structures to achieve a more accurate sampling of conformational space.
- The docking simulations followed by MD were limited to 100 ns. The post-docking MD simulations should be extended to at least 200 ns for a more robust evaluation.
- Statistical Robustness:
- MD simulations should be performed in triplicates with different seed velocities to ensure statistical significance in the observed trends. This would account for variability and enhance the reliability of the reported results.
- Binding Free Energy Calculations:
- The MM-GBSA binding free energy calculations were conducted over a 20 ns timeframe, which is insufficient for robust conclusions. Performing these calculations over the extended simulations of 200–300 ns is recommended as suggested. The frames should be clustered by combining trajectories from all replicates to select representative conformations for binding free energy calculations.
5. MD Simulation methodology
- The authors should provide a complete protocol used for MD simulations. The manuscript didn’t mention which force field was used or how much time the system was equilibrated in NVT and NPT ensembles.
Suggested Future Directions
- Advanced Sampling Techniques:
- Future works could incorporate enhanced sampling methods like metadynamics, replica-exchange molecular dynamics (REMD), or Gaussian-accelerated MD (GaMD) to explore the conformational landscapes of ITGB1, ITGB3, and ITGB7 in greater detail.
Specific Points
- Discussion:
- While the discussion section is comprehensive, it could benefit from a more in-depth comparison with similar studies or previous findings on calpain-cleavage-related mutations in integrins.
Author Response
Rebuttal for Reviewer-2
Thank you very much for taking the time to review this manuscript. Please find the detailed responses below and the corresponding revisions/corrections have been highlighted in yellow in the revised manuscript.
General Comments
Grammar and Spell Check: The manuscript should undergo a thorough grammar and spellcheck to enhance its readability.
Rebuttal: As per the reviewer’s suggestion, we have refined the manuscript carefully to improve its grammar and spelling for better readability using the online tool, Curie. Kindly review the revised manuscript.
Methodological Improvements
Molecular Dynamics (MD) Simulations:
MD simulations were performed for 100 ns, which may not provide adequate conformational sampling for systems such as ITGB1, ITGB3, and ITGB7, which are crucial to this study. I strongly recommend extending the simulations to 300–400 ns for wild-type and mutant structures to achieve a more accurate sampling of conformational space.
The docking simulations followed by MD were limited to 100 ns. The post-docking MD simulations should be extended to at least 200 ns for a more robust evaluation.
Rebuttal: The optimal time for MD simulations varies based on the complexity of the system and the research question being addressed. In the proposed study, the impact of the point mutations in ITGB1, ITGB3 and ITGB7 on the structural fluctuations was evaluated in comparison to the wild-type proteins. To address this, MD simulations were performed for a period of 100 ns and the structural fluctuations were assessed by calculating the RMSD, RMSF and radius of gyration which is sufficient enough to address the structural fluctuations during the initial signalling period of a receptor-ligand interaction. Since Integrin is a transmembrane receptor and also a calpain substrate, a point mutation in its sequence may impact its interaction with calpain and further downstream signalling pathways mediated by the calpain-integrin axis. Previous studies which addressed the structural impact of pathogenic point mutations in human proteins and their interaction network have performed the MD simulations in the time range of 50 ns -100 ns which are as follows:
MD Simulation of 50 ns
- N N, Zhu H, Liu J, V K, C GPD, Chakraborty C, et al. (2015) Analysing the Effect of Mutation on Protein Function and Discovering Potential Inhibitors of CDK4: Molecular Modelling and Dynamics Studies. PLoS ONE 10(8): e0133969.https://doi.org/10.1371/journal.pone.0133969
- Borgohain G, Dan N, Paul S. Use of molecular dynamics simulation to explore structural facets of human prion protein with pathogenic mutations. Biophys Chem. 2016 Jun;213:32-9. doi: 10.1016/j.bpc.2016.03.004.
MD simulation of 60 ns:
- Sang P, Xie YH, Li LH, Ye YJ, Hu W, Wang J, Wan W, Li R, Li LJ, Ma LL, Li Z, Liu SQ, Meng ZH. Effect of the R119G mutation on human P5CR structure and its interactions with NAD: Insights derived from molecular dynamics simulation and free energy analysis. Comput Biol Chem. 2017 Apr;67:141-149. doi: 10.1016/j.compbiolchem.2016.12.015.
MD Simulation of 100 ns:
- Rucker G, Qin H, Zhang L. Structure, dynamics and free energy studies on the effect of point mutations on SARS-CoV-2 spike protein binding with ACE2 receptor. PLoS One. 2023 Oct 5;18(10):e0289432. doi: 10.1371/journal.pone.0289432.
From our study with 100 ns MD simulation, structural fluctuations in the mutant structures are evident from the RMSD, RMSF and radius of gyration values when compared to the wild-type proteins. These fluctuations may further impact the downstream signalling pathway mediated by the calpain-integrin axis.
We appreciate the reviewer's valuable suggestion regarding extending the MD simulations to 100 ns to capture potential structural fluctuations during the later stages of receptor-ligand interaction. Such analyses could indeed provide additional insights into the impact of the point mutations. However, we would also like to inform you that the interpretations from this molecular modelling study are currently being validated through wet lab experiments. These experiments utilize purified integrins and calpains, employing protein-protein interaction tools and enzyme assays to corroborate our findings. We would be delighted to submit the results of this validation study to your esteemed journal as a follow-up to further validate the conclusions drawn in the current work.
Statistical Robustness:
MD simulations should be performed in triplicates with different seed velocities to ensure statistical significance in the observed trends. This would account for variability and enhance the reliability of the reported results.
Rebuttal: Thank you for your valuable suggestion. However, performing MD simulations in triplicates with different seed velocities in our current setup would require a substantial time investment of approximately 4–6 months. Given that we are already engaged in validating the in-silico findings through comprehensive wet lab experiments, we believe that repeating these simulations would not significantly enhance the conclusions of this in-silico study.
Additionally, we would like to highlight similar studies in the literature where MD simulations were performed as single runs rather than in replicates. This approach has been widely accepted and has provided meaningful insights, further supporting the methodology employed in our study.
- Pestana-Nobles R, Aranguren-Díaz Y, Machado-Sierra E, Yosa J, Galan-Freyle NJ, Sepulveda-Montaño LX, Kuroda DG, Pacheco-Londoño LC. Docking and Molecular Dynamic of Microalgae Compounds as Potential Inhibitors of Beta-Lactamase. Int J Mol Sci. 2022 Jan 31;23(3):1630. doi: 10.3390/ijms23031630
- Yu Y, Wang Z, Wang L, Tian S, Hou T, Sun H. Predicting the mutation effects of protein-ligand interactions via end-point binding free energy calculations: strategies and analyses. J Cheminform. 2022 Aug 20;14(1):56. doi: 10.1186/s13321-022-00639-y.
- Sang P, Xie YH, Li LH, Ye YJ, Hu W, Wang J, Wan W, Li R, Li LJ, Ma LL, Li Z, Liu SQ, Meng ZH. Effect of the R119G mutation on human P5CR structure and its interactions with NAD: Insights derived from molecular dynamics simulation and free energy analysis. Comput Biol Chem. 2017 Apr;67:141-149. doi: 10.1016/j.compbiolchem.2016.12.015.
- Singh B, Bulusu G, Mitra A. Effects of point mutations on the thermostability of B. subtilis lipase: investigating non additivity. J Comput Aided Mol Des. 2016 Oct;30(10):899-916. doi: 10.1007/s10822-016-9978-0
Binding Free Energy Calculations:
The MM-GBSA binding free energy calculations were conducted over a 20 ns time frame, which is insufficient for robust conclusions. Performing these calculations over the extended simulations of 200–300 ns is recommended as suggested. The frames should be clustered by combining trajectories from all replicates to select representative conformations for binding free energy calculations.
Rebuttal: Thank you for your valuable feedback and suggestion to extend MM-GBSA binding free energy calculations over 200–300 ns with trajectory clustering from all replicates.
We would like to clarify that our MM-GBSA results, provide a clear and robust comparative analysis of energy values between the wild-type and mutant structures. The last 20 ns of the 100 ns simulation showed stable conformations, which were utilized for the MM-GBSA calculation. The observed energy trends during this period are consistent and sufficient to draw meaningful conclusions regarding the differential binding affinities. Given that we are already engaged in validating the in-silico findings through comprehensive wet lab experiments, we believe that extending these simulations or recalculating the binding free energy would not significantly enhance the conclusions of this in-silico study.
MD Simulation methodology
The authors should provide a complete protocol used for MD simulations. The manuscript didn’t mention which force field was used or how much time the system was equilibrated in NVT and NPT ensembles.
Rebuttal: As per reviewer’s suggestion, we have provided details of MD simulations in the Materials and Methods section (Line 807-813) as follows:
Each protein was solvated in a cubic water box with periodic boundary conditions and ions were added to neutralize the system. The CHARMM27 force field was used for the protein and the TIP3P water model for the solvent. The system was energy minimized to remove any bad contacts or steric clashes. The system was equilibrated in two phases: NVT (constant Number, Volume, Temperature) for 1 ns and NPT (constant Number, Pressure, Temperature) for 1 ns ps. A 100 ns MD simulation was performed for each protein at 310 K using the NPT ensemble.
Suggested Future Directions
Advanced Sampling Techniques:
Future works could incorporate enhanced sampling methods like meta-dynamics, replica-exchange molecular dynamics (REMD), or Gaussian-accelerated MD (GaMD) to explore the conformational landscapes of ITGB1, ITGB3, and ITGB7 in greater detail.
Rebuttal: Thank you for your valuable suggestion. We would be happy to consider these methodologies in future studies to further explore the conformational trajectories of the wild-type and mutant proteins in greater depth, enabling a more comprehensive understanding of their structural dynamics.
Specific Points
Discussion:
While the discussion section is comprehensive, it could benefit from a more in-depth comparison with similar studies or previous findings on calpain-cleavage-related mutations in integrin.
Rebuttal: As per the reviewer’s suggestion, we have discussed the results in more detail as follows and also highlighted in yellow in the revised manuscript.
The acquisition and accumulation of genetic mutations is a hallmark of all cancer types. Studies have shown that each cancer type is associated with a signature mutational pattern (Alexandrov et al., 2020). Therefore, for effective cancer management and to prolong the life expectancy of cancer patients, it is essential to understand the mutational pattern specific to each cancer type to delineate the spatiotemporal dynamics of cancer pathophysiology (Line 621-626).
Other similar studies have shown the impact of pathogenic point mutations on the sensitivity of substrates to proteases. One of such studies reported the difference in the sensitivity of p53 mutants towards calpain with some mutants depicting low sensitivity while other mutants were highly sensitive (Pariat et al., 1997). Some point mutations have been shown to impact the thermostability of the enzyme such as Lipase (Singh et al., 2016). Recently, several studies have addressed the impact of point mutations on the structure of proteins and their physiological interactions to evaluate how these mutations contribute to the disease pathogenesis. For instance, point mutations in the spike protein of the SARS-CoV-2 virus were shown to either strengthen or weaken its interaction with ACE-2 receptor 9 (Rucker et al., 2023). In cancers, the oncogenic point mutations affect the binding kinetics of the protein-protein interaction network causing dysfunction of the intrinsic apoptotic pathway (Zhao et al., 2015). In the case of Cutis Laxa, an inherited disorder of the connective tissue, a point mutation R119G in the enzyme Pyr-roline-5-carboxylate reductase has been shown to weaken the flexibility of the enzyme near the catalytic pocket. This in turn impairs the binding of the cofactor Nicotinamide adenine dinucleotide (NAD) leading to lowered affinity towards the same (Li et al., 2017). Five highly deleterious point mutations (R24C, Y180H, A205T, R210P, and R246C) identified using an integrated computational approach in the Cyclin-Dependent Kinase-4 have been shown to affect its interaction with ligand, flavopiridol (Nagasundaram et al., 2015). Moreover, two point mutations, T193I and R148H in the human prion proteins which are linked to the occurrence of familial Creutz-feld-Jacob disease impact the secondary structure leading to reduced conformational space (Borgohain et al., 2016). Thus, point mutations that are pathogenic or deleterious may impact the structural stability of a protein further impairing the protein-protein interaction and downstream signalling pathway (Line 633-657).
Conditional knockdown of ITGB3 reduces various phenotypes such as cell migration, proliferation, senescence reducing angiogenesis and tumour growth (Kovacheva et al., 2021). Mutations in ITGB1 have been shown to convert benign skin tumours into malignant ones (Ferreira et al., 2009). The cytoplasmic domain of these receptors plays a crucial role in downstream signalling, and mutations in these domains have previously been shown to disrupt various cellular processes. One of the studies, (Fitzpatrick et al., 2014) reported the importance of the C-terminal domain of ITGB1 for interaction with the adapter protein kindlin-2. Similarly, (Pfaff et al., 1999), the cytosolic domain of ITGB3 is crucial for cell adhesion, spreading, endocytosis and phagocytosis (Ylänne et al., 1995). A point mutation in ITGB6 was found to reduce its localization to the focal contacts (Huang et al., 1995) (Line 661-671).
Round 2
Reviewer 1 Report
Comments and Suggestions for Authors
I am still not satisfied with the quality of the figures prepared with PyMol.
First of all , on some of them, the text "for evaluation only" is present, which should not happen on figures published in a good journal.
Image resolution can also be improved. There is an option to export high-resolution images using the command "ray."
Author Response
Comments and Suggestions for Authors
- I am still not satisfied with the quality of the figures prepared with PyMol.First of all, on some of them, the text "for evaluation only" is present, which should not happen on figures published in a good journal. Image resolution can also be improved. There is an option to export high-resolution images using the command "ray."
Rebuttal:
Thank you for the valuable comment. As per the reviewer’s comment, for Figure 5, Figure 6, and Figure 7, high-resolution images have been extracted from the Pymol edu version using the ray command as suggested.
Ray trace modality was set using the command set ray_trace_mode, 0 for natural colors and running the ray command for rendering the image.
Figure 5: (a) Docked complex of wild-type ITGB1 (red) and calpain-2 (cyan); (b) Docked complex of the ITGB1 mutant (T777M) (red) and calpain-2 (cyan). Blue spheres depict the mutation and position. Green spheres depict the calcium bound to calpain-2
Figure 6: (a) Docked complex of wild-type ITGB7 (red) and calpain-2 (cyan); (b) docked complex of the ITGB7 mutant (R760H) (red) and calpain-2 (cyan). Blue spheres depict the mutation and position. Green spheres depict the calcium bound to calpain-2
Figure 7: (a) Docked complex of wild-type ITGB3 (red) and calpain-1 (cyan); (b) docked complex of the ITGB3 mutant (D749N) (red) and calpain-1 (cyan); (c) docked complex of the ITGB3 mutant (R786Q) (red) and calpain-1 (cyan). Blue spheres depict the mutation and position. Green spheres depict the calcium bound to calpain-1
Additionally, Figure 2, Figure 3, Figure 4, Figure 8, Figure 9, and Figure 10 have also been revised for improved image quality.
Figure 2: (a) RMSD of ITGB1 wild-type (black) and mutant (T777M) (red), (b) RMSF of ITGB1 wild-type (black) and mutant (T777M) (red), (c) Rg of ITGB1 wild-type(black) and mutant (T777M) (red)
Figure 3: (a) RMSD of wild-type (black) and mutant ITGB7 R760H (red) (b) RMSF of wild-type (black) and mutant ITGB7 R760H (red) (c) Rg of wild-type (black) and mutant ITGB7 R760H (red)
Figure 4: (a) RMSD of wild-type (black) and mutant ITGB3 (D749N) (red) and (R786Q) (green), (b) RMSF of wild-type (black) and mutant ITGB3 (D749N) (red) and (R786Q) (green), and (c) Rg of wild-type (black) and mutant ITGB3 (D749N) (red) and (R786Q) (green).
Figure 8: (a) RMSD of the docked complex of wild-type (black) and mutant ITGB1(T777M) (red); (b) RMSF of the docked complex of wild-type (black) with calpain-2/m-calpain (red) and mutant ITGB1 (T777M) (green) with calpain-2/m-calpain (blue); (c) Rg of the docked complex of wild-type (black) and mutant ITGB1(T777M) (red); (d) Hydrogen bond of the docked complex of wild-type (black) and mutant ITGB1 (T777M) (red)
Figure 9: (a) RMSD of the docked complex of wild-type (black) and mutant ITGB7 (R760H) (red); (b) RMSF of the docked complex of wild-type (black) with calpain-2/m-calpain (red) and mutant ITGB7 (R760H) (green) with calpain-2/m-calpain (blue); (c) Rg of the docked complex of wild-type (black) and mutant ITGB7(R760H) (red); (d) Hydrogen bond of the docked complex of wild-type (black) and mutant ITGB7 (R760H) (red)
Figure 10: (a) RMSD of the docked complex of wild-type (black) and mutant ITGB3 (D749N) (red) (R786Q) (green); (b) RMSF of the docked complex of wild-type (black) with calpain-1/mu-calpain (red)and mutant ITGB3 (D749N) (green) with calpain-2/m-calpain (blue), (R786Q)(yellow) with calpain-2/m-calpain (brown); (c) Rg of the docked complex of wild-type (black) and mutant ITGB3 (D749N) (red) (R786Q) (green); (d) hydrogen bond of the docked complex of wild-type (black) and mutant ITGB3 (D749N) (red) (R786Q) (green)

Reviewer 2 Report
Comments and Suggestions for Authors
The authors have adequately addressed this reviewer's concerns and I recommend inclusion of this manuscript in its current form. This reviewer is satisfied with the revised version of the manuscript.
Author Response
Comments and Suggestions for Authors
The authors have adequately addressed this reviewer's concerns and I recommend the inclusion of this manuscript in its current form. This reviewer is satisfied with the revised version of the manuscript.
Response: We thank the reviewer for the valuable input to the manuscript and words of encouragement.